# Efficacy and safety of oral Chinese medicine on cancer-related fatigue for lung cancer patients after chemotherapy: Protocol for systematic review and meta-analysis

Peijin Li[1,2☯], Qian Wang[3☯], Lixing Liu[1], Rui Zhou[1,2], Tingting Liu[1,2], Yue Wang[4], Li Feng[1]*

1 Department of Traditional Chinese Medicine/National Cancer Center/National Clinical Research Center for Cancer/Cancer Hospital, Chinese Academy for Medical Sciences and Peking Union Medical College, Beijing, China, 2 Department of Oncology/Dongzhimen Hospital, Beijing University of Chinese Medicine, Beijing, China, 3 Department of Vascular Surgery/Dongfang Hospital, Beijing University of Chinese Medicine, Beijing, China, 4 Department of Oncology/Beijing Hospital of Traditional Chinese Medicine, Beijing, China

☯ These authors contributed equally to this work.
* fengli663@126.com

**Data Availability Statement:** All relevant data from this study are included in our paper and Supporting information.

**Funding:** LF, the Capital Health Research and Development of Special(Grant Number 2020-2-4026)http://wjw.beijing.gov.cn/sy_20013/; LF and

## Abstract

### Introduction

Lung cancer has the highest mortality rate of about 18.0% among malignant tumors worldwide, and chemotherapy is the main treatment. 80% of patients receiving chemotherapy suffers from cancer-related fatigue, which is the most severe symptom, with a large effect on quality of life as well as prognosis. Oral Chinese medicine, a kind of complementary and alternative medicine, has been proved to benefit lung cancer patients. However, no studies have reviewed whether it can reduce fatigue in lung cancer patients after chemotherapy, which is the purpose of our study.

### Methods

Two reviewers will systematically and independently retrieve papers, select studies for inclusion, extract data, and assess risk of bias. The following nine databases will be searched: China National Knowledge Infrastructure, Wan Fang database, Chinese Scientific Journals Database, Chinese biomedical literature service system, PubMed, Web of Science, OVID, Scopus, and EMBASE from inception to February, 2022. Included studies will only be randomized controlled trials. Primary outcome is cancer-related fatigue. Secondary outcomes are quality of life, immunologic function, and the incidence of adverse events. We will use RoB 2 tool to assess the risk of bias and RevMan to analyze data. Risk ratios will be calculated for dichotomous data and mean differences for continuous data. Random-effect model will be used to integrate statistical effects. Meta-regression, subgroup and sensitivity analyses will be carried out. We will evaluate the strength and overall quality of evidence with four levels: very low, low, moderate, and high.

LXL, the National Natural Science Foundation of China(Grant Number 81873283, 82104958)http://www.nsfc.gov.cn/. The funders had and will not have a role in study design, data collection and analysis, decision to publish, or preparation of the manuscript.

## Results

The review of current evidence of oral Chinese medicine on cancer-related fatigue for lung cancer patients after chemotherapy will be narratively summarized and quantitatively analyzed.

## Conclusion

The definitive conclusion will help physicians to determine whether oral Chinese medicine is an effective treatment for reducing fatigue in lung cancer patients after chemotherapy in clinical settings.

## Systematic review registration

PROSPERO CRD42021292576.

## Introduction

Lung cancer (LC) has the highest mortality rate among malignant tumors worldwide and seriously harms public health. According to the latest global research, new cases and mortality due to LC reached 2.2 million and 1.79 million in 2020, accounting for 11.4% and 18.0% of the total number of new cases and deaths due to cancer, respectively [1, 2]. While LC is clinically treated via numerous therapeutic modalities, chemotherapy remains the cornerstone of treatment, and is widely used in neoadjuvant, adjuvant, and metastatic clinical settings, particularly for those with negative gene mutations or less expression of PD-1/L1 [3, 4]. Chemotherapy is associated with multiple adverse effects. Among them, fatigue, also known as cancer-related fatigue (CRF), is perceived as the most severe and distressing symptom, with a large effect on physical and emotional health, cognitive function, social relations, and quality of life (QOL) [5, 6]. CRF can last for several months to years, even more than ten years in 25% of cancer survivors [7]. Reports have shown that the prevalence of CRF in patients with LC is as high as 96%, and the incidence in patients receiving chemotherapy is 80%, contributing to the discontinuation of cancer treatment and poor prognosis [8, 9]. The mechanisms leading to CRF are complicated and largely unknown, but some possibilities have been proposed, such as the dysfunction of skeletal muscle, and the dysregulation of inflammatory cytokines [10]. Studies have revealed platinum-based chemotherapeutic drugs may result in CRF by increasing IL-8 levels [11]. Some evidence suggests the recommendation of physical activities and psychological therapies as a supportive strategy for reducing CRF [12, 13]. However, in addition to the nonpharmacologic interventions mentioned above, there is little evidence regarding medication for CRF treatment, which limits the treatment options in clinical settings. Complementary and alternative medicine has experienced increased popularity owing to its ever-increasing integrative role in the treatment of diseases. Substantial evidence indicates that the treatment, such as Chinese herbal medicine, yoga, tai chi, and acupuncture, can significantly alleviate CRF [14–16]. Among these, Chinese herbal medicine is considered an important therapeutic strategy that has been proved to improve QOL and survival time in LC patients [17, 18]. Meta-analyses have assessed the effects of traditional Chinese medicine (TCM) injection and Chinese herbal medicine on CRF [19, 20]; however, the results regarding the efficacy of oral Chinese medicine (OCM) on fatigue in patients with LC after chemotherapy were unclear. Unfortunately, no studies have reviewed this. Hence, there is a strong need for the

accumulation of current evidence. The purpose of this article is to determine the efficacy and safety of OCM and whether CRF patients with LC benefit from it after chemotherapy. Our study will assist in determining effective interventions based on evidence-based medicine, by building effective, safe and individualized therapeutic strategies.

## Methods

### Study registration

This protocol has been registered on PROSPERO with the number CRD42021292576 (https://www.crd.york.ac.uk/prospero/). It will adhere to the Cochrane Handbook [21] and the Preferred Reporting Items for Systematic Reviews and Meta-analysis Protocols (PRISMA-P) guidelines (S1 Table) [22].

### Eligibility criteria

**Types of studies.** Included studies in this review will only be randomized controlled trials, irrespective of language, publication year, and publication status. Quasi-randomized trials, cases reports, and cross-sectional studies will be excluded.

**Types of participants.** CRF participants, who were diagnosed with LC and received chemotherapy treatment will be included, regardless of pathological type and stage. Patients diagnosed with other types of cancer will be excluded. Studies without stating diagnostic criteria of CRF will be excluded, but there will be no limits on specific diagnostic approaches.

**Types of interventions.** Included studies used OCM as experimental interventions, compared against any control interventions of the followings: placebo, no treatment, conventional treatment, or usual care. There will be no restriction on the types of OCM, such as single Chinese medicine, Chinese patent medicine or Chinese herbal decoction based on syndrome differentiation. If studies used OCM combination with other treatments as experimental interventions, the other treatments in the experimental group should be the same as the control group. Studies involving topical Chinese medicine or TCM injection will be excluded.

**Types of outcomes.** CRF is the primary outcome, which can be measured by the following scales: the Brief Fatigue Inventory (BFI), the Functional Assessment of Chronic Illness Therapy-Fatigue (FACIT-F), the Multidimensional Fatigue Symptom Inventory (MFSI), the Piper Fatigue Scale (PFS), or other validated scales (S2 Table) [23]. Secondary outcomes: (a) QOL, measured by any validated scale, for example, the Functional Assessment of Cancer Therapy-General/Lung (FACT-G/L) [24], or the European Organization for Research and Treatment of Cancer Quality of Life Questionnaire-Core 30/Lung Cancer 43 (EORTC QLQ-C30/LC43) [25]; (b) immunologic function, measured by serum T lymphocyte subsets, such as the ratio of CD4+/CD8+T lymphocytes; (c) the incidence of adverse events, such as liver injury, kidney injury, nausea, vomiting, and constipation, among others. There will be no limits on timing of outcome assessment.

### Search strategy for the identification of studies

Two reviewers (RZ and TTL) will systematically and independently search studies published in the following nine databases: China National Knowledge Infrastructure (CNKI), Wan Fang database, Chinese biomedical literature service system (SinoMed), Chinese Scientific Journals Database (VIP), PubMed, Web of Science, OVID, Scopus, and EMBASE from inception to February, 2022. There will be no limitation based on language. We will additionally search ongoing trials and unpublished trials in the World Health Organization (WHO) International Clinical Trials Registry Platform (ICTRP), the Cochrane Central Register of Controlled Trials

(CENTRAL, The Cochrane Library), and the ClinicalTrials.gov. We will manually search all references of included studies and grey literatures through the OpenGrey database to ensure that no pertinent studies will be omitted. We will use the following MeSH terms and keywords: fatigue, cancer related fatigue, CRF, chemotherapy-induced fatigue, chemotherapy-related fatigue, lung neoplasm, lung cancer, pulmonary neoplasm, pulmonary cancer, traditional Chinese medicine, Chinese medicine, Chinese herbal drugs, herbal medicine, herb, randomized controlled trial, controlled clinical trial, randomized, drug therapy, and placebo. The complete search strategy for PubMed is shown in Table 1 and other strategies can be found in S1 File. Any inconsistencies will be solved by discussion and consensus with a third reviewer (LXL).

## Data collection and analysis

**Selection of studies.** After retrieving studies based on search strategies and removing duplicates, two reviewers (RZ and TTL) will independently screen the titles and abstracts for the first inclusion and the full texts for the final inclusion. EndnoteX9 will be utilized to screen the titles and abstracts. Studies that are clearly inconsistent with the inclusion criteria will be rejected, and the reasons for any excluding one will be recorded. Any inconsistency will be consulted and solved by a third investigator (LXL). The research flowchart recommended by the PRISMA statement is shown in Fig 1.

**Data extraction and management.** Essential data will be extracted into an excel spreadsheet independently by two reviewers (YW and RZ) from included studies. When they can't reach a consensus, the third reviewer (QW) should make a judgment. Required information is as follows: reference ID, author names, publication year, study design, sample size, basic information of participants (age, sex, smoking history, pathological type and stage of LC, and previous treatment), OCM intervention and comparator (randomization, allocation, blinding, types and duration of intervention, and details of OCM), follow-up period, types of outcomes measures, and other outcomes. If necessary, the authors will be contacted to acquire certain information that can not be found in the studies. The data after double-checking will be recorded to the Review Manager V.5.3 by two reviewers (YW and RZ) independently.

**Assessment of risk of bias.** Two reviewers (RZ and LXL) will separately assess risk of bias (low risk, high risk, or some concerns) using RoB 2 tool [26], the revised Cochrane risk-of-bias tool, including 5 aspects: randomization process (for example, whether the sequence generation is random, and the allocation is concealed), deviations from intended interventions (such as blinding), missing outcomes (for example, whether the outcome data is complete), outcome measurement (for example, whether measures are appropriate), and selection report (for example, whether the results are selective reported). Any discrepancy will be resolved by consensus with a third reviewer (PJL).

**Assessment of publication bias.** Potential publication bias will be analyzed and shown by the funnel plot if no less than ten studies can be used for meta-analysis. Egger's test will also be used to assess publication bias, and P values below 0.05 indicate large such bias.

**Measures of treatment effect.** Data will be analyzed through RevMan V.5.3 software. We will calculate risk ratios (RR) and 95% confidence intervals for dichotomous variables, as well as weighted mean difference (WMD) or standard mean differences (SMD) and 95% confidence intervals for continuous data. Standard mean differences will be adopted when included studies use different measure tools.

**Dealing with missing data.** Two reviewers (PJL and QW) will be responsible for contacting the corresponding author through e-mails or phones to obtain the elusive or missing data. The study will be excluded if the data of which cannot be obtained. The potential influence of missing data on the analysis will be measured by a sensitivity analysis.

**Table 1. Search strategy for PubMed.**

| NO. | Search Terms |
|---|---|
| #1 | fatigue [MeSH Terms] |
| #2 | cancer related fatigue |
| #3 | CRF |
| #4 | Fatigue syndrome, chronic [MeSH Terms] |
| #5 | chemotherapy-induced fatigue |
| #6 | chemotherapy-related fatigue |
| #7 | #1-#6/OR |
| #8 | Lung Neoplasms [MeSH Terms] |
| #9 | Lung Neoplasm |
| #10 | Neoplasms, Lung |
| #11 | Neoplasm, Lung |
| #12 | Pulmonary Neoplasms |
| #13 | Pulmonary Neoplasm |
| #14 | Neoplasms, Pulmonary |
| #15 | Neoplasm, Pulmonary |
| #16 | Lung Cancer |
| #17 | Lung Cancers |
| #18 | Cancer, Lung |
| #19 | Cancers, Lung |
| #20 | Cancer of Lung |
| #21 | Cancer of the Lung |
| #22 | Pulmonary Cancer |
| #23 | Pulmonary Cancers |
| #24 | Cancer, Pulmonary |
| #25 | Cancers, Pulmonary |
| #26 | Small Cell Lung Carcinoma [MeSH Terms] |
| #27 | Carcinoma, Non-Small-Cell Lung [MeSH Terms] |
| #28 | #8-#27/OR] |
| #29 | Medicine, Chinese traditional [MeSH Terms] |
| #30 | Traditional Medicine, Chinese |
| #31 | Traditional Chinese Medicine |
| #32 | Traditional Chinese Medicines |
| #33 | Chinese Traditional Medicine |
| #34 | Chinese medicines |
| #35 | Chinese medicine |
| #36 | Chinese Medicine, Traditional |
| #37 | Drugs, Chinese Herbal |
| #38 | Herb |
| #39 | Herbs |
| #40 | Herbal Medicine [MeSH Terms] |
| #41 | Herbal Medicines |
| #42 | Medicine, Herbal |
| #43 | Chinese Herbal Drugs |
| #44 | Herbal Drugs, Chinese |
| #45 | #29-#44/OR |
| #46 | groups [Title/Abstract] |
| #47 | trial [Title/Abstract] |

(*Continued*)

**Table 1.** (Continued)

| NO. | Search Terms |
|---|---|
| #48 | controlled clinical trial [Publication Type] |
| #49 | randomized controlled trial [Publication Type] |
| #50 | randomly [Title/Abstract] |
| #51 | randomized [Title/Abstract] |
| #52 | placebo [Title/Abstract] |
| #53 | drug therapy [MeSH Subheading] |
| #54 | #46-#53/OR |
| #55 | animals [MeSH Terms] NOT humans [MeSH Terms] |
| #56 | #54 NOT #55 |
| #57 | #7 AND #28 AND #45 AND #56 |

**Assessment of heterogeneity.** Statistical heterogeneity will be analyzed by Q test, significant at P-value < 0.10. The level of heterogeneity will be described by the $I^2$ statistic, and values of 0%, 25%, 50%, and 75% are considered to represent no, mild, moderate, and extreme heterogeneity, respectively. We will use random-effect model to integrate statistical effects because of anticipated heterogeneity. Potential sources of heterogeneity will be explored through meta-regression and subgroup analyses. If considerable heterogeneous exists across studies, meta-analysis should not be conducted, and a narrative report will be provided.

**Subgroup analysis.** If included studies are adequate, subgroup analysis will be conducted in RevMan V.5.3 based on pathological type and stage of LC, chemotherapy regimen, OCM alone or combination with other treatments, type and duration of intervention, follow-up period, and type of outcome measure.

**Sensitivity analysis.** We will use a sensitivity analysis to evaluate whether the pooled results are reliable. The studies with low quality will be excluded, and the pooled effect size will

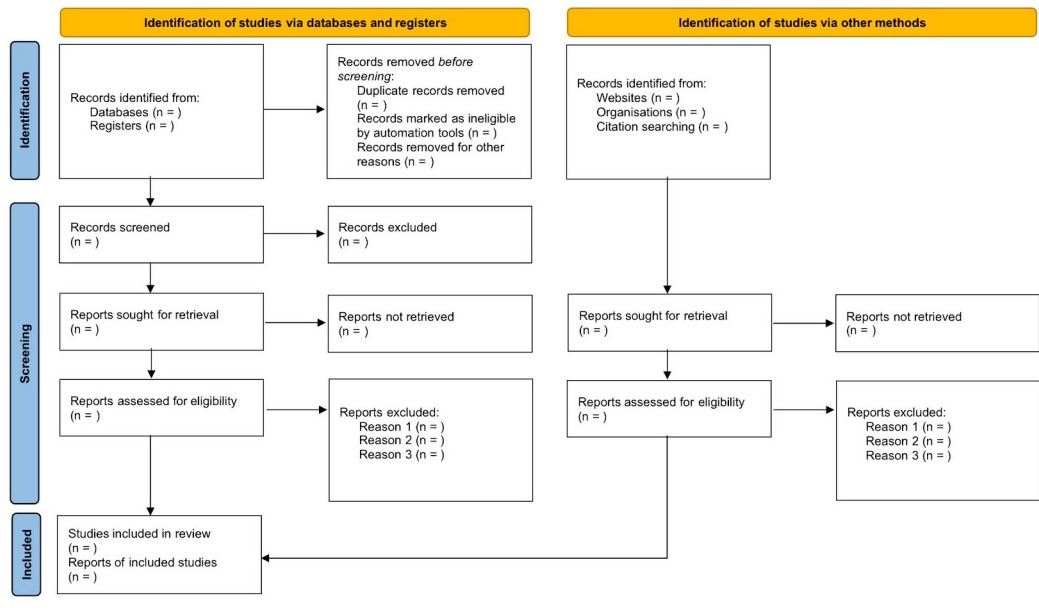

**Fig 1. PRISMA flowchart of studies selection.**

be analyzed again to see whether the outcomes change significantly. In addition, sensitivity analyses will be performed by using fixed-effect model and removing one study each time.

**Grading the quality of evidence.**   We will evaluate the strength of evidence (very low, low, moderate, and high) using The Grading of Recommendations Assessment, Development, and Evaluation (GRADE) tool [27], including 5 major aspects: risk of bias, publication bias, imprecision, inconsistency, and indirectness.

**Ethics and dissemination.**   This study does not need ethical approval, since it is a protocol for systematic review. Also, no privacy data of participants will be involved, and no intervention will be performed on them. The final analysis will be disseminated through the PROSPERO website and peer-reviewed journals. Academic conferences will also share it if needed.

**Patient and public involvement.**   This study is designed without patient and public involvement as well as performance.

**Amendments.**   This report will involve the following major stages: search and selection of studies, extraction and management of data, and data analysis. Any changes in this original protocol for systematic review will be updated on the PROSPERO website and stated formally in the final review manuscript. If necessary, the date and reason for any changes will also be provided.

## Discussion

LC remains the leading cause of cancer-related deaths worldwide and is associated with significant morbidity and mortality [1, 2]. Chemotherapy is a major treatment for LC. Reports have shown that, after chemotherapy, half of LC patients experience moderate to severe CRF, lasting for several months or longer, which seriously affects patients QOL [28]. CRF is is characterized by tiredness and many patients have difficult tolerating CRF, even more so than other symptoms such as, pain, diarrhea, decreased appetite, nausea, and vomiting. However, there are currently few drug treatments available for CRF other than some nonpharmacological interventions, such as physical activities and psychological therapies, leading to treatment limitations. Additional strategies are needed to increase CRF treatment. TCM technical committee was created in 2009 [29], showing that TCM is more widely used worldwide. OCM, an essential part of TCM, reportedly improves the overall survival of LC patients, suppresses the development of LC, reduces treatment-related adverse effects, and relieves symptoms of fatigue [30–33]. Therefore, it is important to develop evidence-based OCM strategies for CRF treatment. Given that no reviews have evaluated the efficacy and safety of OCM on CRF in patients with LC after chemotherapy, we will conduct an objective and comprehensive review of the existing evidence. We hope that this report will reach a definitive conclusion and help physicians determine whether OCM is an effective treatment for reducing CRF in LC patients after chemotherapy in clinical settings.

## Supporting information

**S1 Table. PRISMA-P 2015 checklist.**
(PDF)

**S2 Table. Measures of CRF.**
(PDF)

**S1 File. Search strategies.**
(PDF)

## Author Contributions

**Conceptualization:** Peijin Li, Qian Wang.

**Data curation:** Peijin Li, Qian Wang, Lixing Liu, Rui Zhou, Tingting Liu, Yue Wang.

**Formal analysis:** Peijin Li, Qian Wang, Lixing Liu, Rui Zhou.

**Funding acquisition:** Lixing Liu, Li Feng.

**Investigation:** Lixing Liu, Rui Zhou, Tingting Liu.

**Methodology:** Lixing Liu, Rui Zhou.

**Supervision:** Peijin Li, Qian Wang, Li Feng.

**Validation:** Lixing Liu, Tingting Liu, Yue Wang.

**Writing – original draft:** Peijin Li, Qian Wang.

**Writing – review & editing:** Lixing Liu, Rui Zhou, Tingting Liu, Yue Wang, Li Feng.

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
