## [Decision Letter · Decision Letter 0]

13 Apr 2022

PONE-D-21-40607Efficacy and safety of oral Chinese medicine on cancer-related fatigue for lung cancer patients after chemotherapy: Protocol for systematic review and meta-analysisPLOS ONE

Dear Dr. Li,

Thank you for submitting your manuscript to PLOS ONE. After careful consideration, we feel that it has merit but does not fully meet PLOS ONE’s publication criteria as it currently stands. Therefore, we invite you to submit a revised version of the manuscript that addresses the points raised during the review process.

We look forward to receiving your revised manuscript.

Kind regards,

Ning Wei

Academic Editor

PLOS ONE

Journal Requirements:

Additional Editor Comments (if provided):

Please address all the concerns of reviewers.

Reviewers' comments:

Reviewer's Responses to Questions

**Comments to the Author**

1. Does the manuscript provide a valid rationale for the proposed study, with clearly identified and justified research questions?

Reviewer #1: Yes

Reviewer #2: Partly

2. Is the protocol technically sound and planned in a manner that will lead to a meaningful outcome and allow testing the stated hypotheses?

Reviewer #1: Partly

Reviewer #2: Partly

3. Is the methodology feasible and described in sufficient detail to allow the work to be replicable?

Reviewer #1: Yes

Reviewer #2: Yes

4. Have the authors described where all data underlying the findings will be made available when the study is complete?

Reviewer #1: Yes

Reviewer #2: Yes

5. Is the manuscript presented in an intelligible fashion and written in standard English?

Reviewer #1: No

Reviewer #2: No

6. Review Comments to the Author

You may also provide optional suggestions and comments to authors that they might find helpful in planning their study.

Reviewer #1: In this paper, the authors tried to set up a protocol for systematic review and meta-analysis in efficacy and safety of oral Chinese medicine on cancer-related fatigue for lung cancer patients after chemotherapy. They used two parts to explain their methods, which may be useful for physicians to decide whether oral Chinese medicine can be used as an adjunctive treatment. The protocol was described in detail, but some issues may need to be explained.

1. Chinese medicine is mainly used by Chinese, so the availability of this analysis can be limited. Do the authors think it may also applicable to other countries?

2. Please correct the spelling mistakes (such as Line 68, 106), punctuations (such as Line 75, 125) and reorganization of sentences.

3. For S1 table, Item No 16 “Specify any planned assessment of meta-bias(es) (such as publication bias across studies, selective reporting within studies”, did not report on P14.

4. The Introduction is long and not so specific, the authors should rewrite this part in a better way.

Reviewer #2: Li et al. described a protocol for a meta-analysis to determine the efficacy and safety of oral Chinese medicine and whether cancer-related fatigue patients with lung cancer after chemotherapy can benefit from it.

1. The authors seem only to want to evaluate the effects only after chemotherapy treatment. Whether the team will consider those who were treated with surgical resection, immunotherapy, or targets therapy? If only chemotherapy will be included, whether ‘chemotherapy-related fatigue’ is more suitable?

2. Please cite the most recent version of Cochrane handbook.

3. Timing of outcome assessment should be clarified.

4. In the search strategies section, please use the RCT filters described in Cochrane handbook and display strategy for each database.

5. Whether oral Chinese medicine could be considered as one entity also raised concern. Clinical heterogeneity exists. I recommend the usage of random-effect model in the future study. Besides, subgroup analysis is needed for this question.

6. I recommend using RoB 2 tool for risk of bias assessment.

7. Whether there are existing meta-analyses concerning this topic. If yes, please explain the reason you are conducting this study in the introduction section

8. PRISMA flowchart should be in accordance with PRISMA 2020.

9. The English language should be edited by a native speaker.

7. PLOS authors have the option to publish the peer review history of their article (what does this mean?). If published, this will include your full peer review and any attached files.

Reviewer #1: No

Reviewer #2: **Yes: **Zhaolun Cai

---

## [Author Response · Author response to Decision Letter 0]

26 May 2022

Dear Editor:

Thank you for your kind letter and for the reviewers’ comments concerning our manuscript entitled “Efficacy and safety of oral Chinese medicine on cancer-related fatigue for lung cancer patients after chemotherapy: Protocol for systematic review and meta-analysis” (ID: PONE-D-21-40607). Those comments are all valuable and very helpful for revising and improving our paper, as well as the important guiding significance to our researches. We have studied comments carefully and have revised the manuscript in accordance with your kind advice and referee’s detailed suggestions, and have carefully proof-read the manuscript to minimize typographical, grammatical, and bibliographical errors. Here below is our description on revision according to the reviewers’ comments:

Reviewer 1:

Comment 1: Chinese medicine is mainly used by Chinese, so the availability of this analysis can be limited. Do the authors think it may also applicable to other countries?

Response: Thanks for the referee’s comment. Chinese medicine, considered as a kind of complementary and alternative medicine, is widely used in Japan, Korea, Germany and other countries. It is recognized across the world that Chinese medicine plays an important role in the treatment of cancer, COVID-19, and other diseases. We believe that this analysis can provide valuable information to physicians in China as well as other countries.

Comment 2: Please correct the spelling mistakes (such as Line 68, 106), punctuations (such as Line 75, 125) and reorganization of sentences.

Response: Thanks for the referee’s suggestion. We are very sorry for our incorrect spellings, punctuations and reorganization of sentences. Multiple checks have been performed, and any mistakes have been corrected in the paper. The English language of full text has been revised by an expert whose first language is English. Also, we have modified and perfected the content of our manuscript in accordance with the referee’s suggestions, especially the abstract, introduction and discussion sections.

Comment 3: For S1 table, Item No 16 “Specify any planned assessment of meta-bias(es) (such as publication bias across studies, selective reporting within studies”, did not report on P14.

Response: Thanks the referee for drawing my attention to a mistake appearing in S1 table. The page Item No 16 reported on has been corrected as “P12-13”. Assessment of risk of bias in 5 aspects (randomization process, deviations from intended interventions, missing outcomes, missing outcomes, and selection report) was reported in Line 206-216, P12-P13, and publication bias was reported in Line 217-221, P13 in our revised manuscript.

Comment 4: The Introduction is long and not so specific, the authors should rewrite this part in a better way.

Response: Thanks for the referee’s kind advice, which is very beneficial to the improvement of our article. According to the proposal, we have simplified and streamlined the Introduction (Line 60-106, P3-P5) so that it is more clearly to present the main background questions and hopefully more to the reviewer’s liking.

Special thanks to you for your good comments.

Reviewer #2:

Comment 1: The authors seem only to want to evaluate the effects only after chemotherapy treatment. Whether the team will consider those who were treated with surgical resection, immunotherapy, or targets therapy? If only chemotherapy will be included, whether ‘chemotherapy-related fatigue’ is more suitable?

Response: Thanks for the referee’s valuable suggestions. This study is focus only on lung cancer patients after chemotherapy treatment, irrespective of surgical resection, immunotherapy, or targets therapy. Cancer-related fatigue is described as a tiredness or exhaustion related to cancer or cancer treatment in NCCN guidelines and ESMO Clinical Practice Guidelines, which is also a candidate term in Emtree, while chemotherapy-related fatigue is not. Therefore, we think cancer-related fatigue is more formal and more widely used. Thanks again for your advice, which is very helpful for us.

Comment 2: Please cite the most recent version of Cochrane handbook.

Response: Thanks very much for the referee’s important suggestion. We have cited the most recent version of Cochrane handbook in Line 112, P6 in our revised manuscript.

Comment 3: Timing of outcome assessment should be clarified.

Response: Thanks for the referee’s kind advice. We will place no limitations on timing of outcome assessment (Line 151-152, P7). We will extract the duration of intervention and follow-up period from included studies (Line 200-201, P12). If possible, we will also conduct subgroup analysis for the duration of intervention and follow-up period (Line 248-249, P14). The above supplements have been added in our revised manuscript.

Comment 4: In the search strategies section, please use the RCT filters described in Cochrane handbook and display strategy for each database.

Response: Thanks for the referee’s suggestion. We have read the use of the RCT filters described in Cochrane handbook carefully and have reviewed the relevant papers. In accordance with the referee’s suggestion, we have refined the search strategy for PubMed (Table 1) and have added the search strategies for other databases in S3 File.

Comment 5: Whether oral Chinese medicine could be considered as one entity also raised concern. Clinical heterogeneity exists. I recommend the usage of random-effect model in the future study. Besides, subgroup analysis is needed for this question.

Response: Thanks for the referee’s kind advice. We believe that the future study will be useful to determine whether oral Chinese medicine can be considered as one entity and an effective treatment for cancer-related fatigue. According to the referee’s proposal, we will use random-effect model to integrate statistical effects in the future study (Line 240-241, P14). Because of anticipated heterogeneity, meta-regression and subgroup analyses will be performed to explore the sources of heterogeneity (Line 241-242, P14). Subgroup analysis for the pathological type and stage of lung cancer, the type and duration of intervention, type of outcome measure, and follow-up period will be conducted in the future study (Line 245-249, P14). 

Comment 6: I recommend using RoB 2 tool for risk of bias assessment.

Response: Thanks for the referee’s suggestion, which is very correct and helpful for our study. RoB 2 tool, the version 2 of the Cochrane tool for assessing risk of bias in randomised trial, is an authoritative tool. We have revised the assessment of risk of bias section in accordance with your recommendation (Line 206-216, P12-13).

Comment 7: Whether there are existing meta-analyses concerning this topic. If yes, please explain the reason you are conducting this study in the introduction section.

Response: Thanks for the referee’s kind advice, which is very beneficial to the improvement of our article. There are several meta-analyses related to this topic, but they are not entirely consistent with our analysis. These meta-analyses assessed the effectiveness and safety of Chinese herbal medicine and traditional Chinese medicine injection for the treatment of cancer-related fatigue, which didn’t focus only on lung cancer patients. No studies have reviewed the efficacy and safety of oral Chinese medicine on fatigue in lung cancer patients after chemotherapy. We have added the relevant content in the introduction section according to the referee’s advice (Line 95-101, P5).

Comment 8: PRISMA flowchart should be in accordance with PRISMA 2020.

Response: Thanks for the referee’s very helpful suggestion. We have updated the PRISMA flowchart in accordance with PRISMA 2020 (Fig 1). This is an omission on our part. Thanks again for the referee’s advice.

Comment 9: The English language should be edited by a native speaker.

Response: Thanks for the referee’s kind advice. The English language has been revised by an expert whose first language is English. We hope that the English language in our revised manuscript meet your expectations.

Special thanks to you for your good comments.

Other changes:

1.We have modified and checked the format of our article to ensure that out manuscript meet PLOS ONE’s style requirements.

2.We have added the data availability in the cover letter, and have provided the minimal underlying data in S3 File.

3.After consulting an expert in methodology, we have added OVID and Scopus databases to search published studies, as well as the ClinicalTrials.gov to search ongoing and unpublished trials. We have also added the OpenGrey database to search grey literatures. These changes will be updated on the PROSPERO website.

We have tried our best to improve the manuscript and have revised many grammatical or typographical errors. All the lines and pages indicated above are in the revised manuscript. All the changes will not influence the content and framework of the paper. And here we did not list all changes but marked in red/yellow in the revised manuscript.

We appreciate for the warm work of Editors and Reviewers earnestly, and hope that the correction will meet with approval. 

Thank you and all the reviewers very much the comments and suggestions.

Sincerely yours,

Peijin Li

---

## [Decision Letter · Decision Letter 1]

7 Jun 2022

Efficacy and safety of oral Chinese medicine on cancer-related fatigue for lung cancer patients after chemotherapy: Protocol for systematic review and meta-analysis

PONE-D-21-40607R1

Dear Dr. Feng,

We’re pleased to inform you that your manuscript has been judged scientifically suitable for publication and will be formally accepted for publication once it meets all outstanding technical requirements.

Kind regards,

Ning Wei

Academic Editor

PLOS ONE

Additional Editor Comments (optional):

The revised version of this manuscript is acceptable.

Reviewers' comments:

Reviewer's Responses to Questions

**Comments to the Author**

1. Does the manuscript provide a valid rationale for the proposed study, with clearly identified and justified research questions?

Reviewer #1: Yes

Reviewer #2: Yes

2. Is the protocol technically sound and planned in a manner that will lead to a meaningful outcome and allow testing the stated hypotheses?

Reviewer #1: Yes

Reviewer #2: Partly

3. Is the methodology feasible and described in sufficient detail to allow the work to be replicable?

Reviewer #1: Yes

Reviewer #2: Yes

4. Have the authors described where all data underlying the findings will be made available when the study is complete?

Reviewer #1: Yes

Reviewer #2: Yes

5. Is the manuscript presented in an intelligible fashion and written in standard English?

Reviewer #1: Yes

Reviewer #2: Yes

6. Review Comments to the Author

You may also provide optional suggestions and comments to authors that they might find helpful in planning their study.

Reviewer #1: In this paper, the authors tried to set up a protocol for systematic review and meta-analysis in efficacy and safety of oral Chinese medicine on cancer-related fatigue for lung cancer patients after chemotherapy. They used two parts to explain their methods, which may be useful for physicians to decide whether oral Chinese medicine can be used as an adjunctive treatment. The protocol was described in detail, only few issues may need to be explained.

1. What is the differences between Chinese herbal medicine and oral Chinese medicine (OCM), they should have some overlaps and how the authors group them? If Chinese herbal medicine also included OCM, whether there are existing meta-analyses concerning this topic?

Reviewer #2: The authors revised the manuscript in accordance with the comments.

7. PLOS authors have the option to publish the peer review history of their article (what does this mean?). If published, this will include your full peer review and any attached files.

Reviewer #1: No

Reviewer #2: **Yes: **Zhaolun Cai

---

## [Editor Report · Acceptance letter]

21 Jun 2022

PONE-D-21-40607R1 

Efficacy and safety of oral Chinese medicine on cancer-related fatigue for lung cancer patients after chemotherapy: Protocol for systematic review and meta-analysis 

Dear Dr. li:

I'm pleased to inform you that your manuscript has been deemed suitable for publication in PLOS ONE. Congratulations! Your manuscript is now with our production department. 

Kind regards, 

on behalf of

Dr. Ning Wei 

Academic Editor

PLOS ONE